# Martian Multichannel Diode Laser Spectrometer (M-DLS) for In-Situ Atmospheric Composition Measurements on Mars Onboard ExoMars-2022 Landing Platform

**Alexander Rodin** [1,2,*], **Imant Vinogradov** [2], **Sergei Zenevich** [1,2,*], **Maxim Spiridonov** [1,2,3], **Iskander Gazizov** [1,2], **Viktor Kazakov** [1,2], **Viacheslav Meshcherinov** [1,2], **Ilya Golovin** [2,4], **Tatyana Kozlova** [2], **Yuri Lebedev** [2], **Svetlana Malashevich** [1,2], **Artem Nosov** [2], **Oksana Roste** [2], **Alla Venkstern** [2], **Artem Klimchuk** [1], **Vladimir Semenov** [1], **Viktor Barke** [2], **Georges Durry** [5], **Mélanie Ghysels-Dubois** [5], **Elena Tepteeva** [2] and **Oleg Korablev** [2]

1   Moscow Institute of Physics and Technology (MIPT), 141701 Dolgoprudny, Russia; max@nsc.gpi.ru (M.S.); gazizov.ish@phystech.edu (I.G.); kazakov44@rambler.ru (V.K.); meshcherinov@phystech.edu (V.M.); sv.malashevich@gmail.com (S.M.); art.klimchuk@gmail.com (A.K.); semenov.v.m@gmail.com (V.S.)

2   Space Research Institute of the Russian Academy of Sciences (IKI RAS), 117997 Moscow, Russia; imant@iki.rssi.ru (I.V.); golovnin@physics.msu.ru (I.G.); kozlova.t@rssi.ru (T.K.); yv.lebedev@yandex.ru (Y.L.); nosov.a@rssi.ru (A.N.); land@iki.rssi.ru (O.R.); brig137@mail.ru (A.V.); tandem422t@mail.ru (V.B.); tepteeva.es@phystech.edu (E.T.); korab@iki.rssi.ru (O.K.)

3   Prokhorov General Physics Institute of the Russian Academy of Sciences (GPI RAS), 119991 Moscow, Russia

4   Faculty of Physics, Lomonosov Moscow State University, 119991 Moscow, Russia

5   Groupe de Spectrométrie Moléculaire et Atmosphérique Unité Mixte de Recherche Centre National de la Recherche Scientifique (GSMA, UMR CNRS) 7331, Université de Reims, BP 1039, CEDEX 2, 51687 Reims, France; georges.durry@univ-reims.fr (G.D.); melanie.ghysels-dubois@univ-reims.fr (M.G.-D.)

*   Correspondence: alexander.rodin@phystech.edu (A.R.); zenevich.sg@phystech.edu (S.Z.); Tel.: +7-498-744-65-24 (A.R. & S.Z.)

**Abstract:** We present a concept of the Martian multichannel diode laser spectrometer (M-DLS) instrument, a part of the science payload onboard Kazachok landing platform in the framework of the ExoMars mission second stage. The instrument, a laser spectrometer operating in the mid-IR spectral range, is aimed at long-term monitoring of isotopic ratios in main Martian volatiles—carbon dioxide and water vapor—in the near-surface atmosphere. The M-DLS spectrometer utilizes the integrated cavity output spectroscopy (ICOS) technique to enhance an effective optical path length and combines high sensitivity and measurement accuracy with relatively simple and robust design. Provided proper compensation of systematic errors by data post-processing, retrievals of main isotopic ratios with relative accuracy of 1% to 3% are expected during at least one Martian year.

**Keywords:** Mars; atmosphere; laser spectroscopy; distributed feedback (DFB) laser; ICOS; water isotopologues; carbon dioxide isotopologues

## 1. Introduction

Martian multichannel diode laser spectrometer (M-DLS) is a laser spectrometer developed by Space Research Institute of Russian Academy of Sciences (IKI RAS) in cooperation with Moscow Institute of Physics and Technology (MIPT) in the framework of the joint Russian–European ExoMars mission. M-DLS is a part of the science payload of the landing platform "Kazachok" to be launched to Mars along with "Rosalind Franklin" rover as a second stage of the ExoMars space mission. The main

purpose of M-DLS is comprehensive study of the Martian atmosphere by continuous measurements of its chemical and isotopic compositions at a fixed location at the planet's surface within on least one Martian year.

　　　Comprehensive study of composition of the Martian atmosphere has been a priority of nearly all mission to the red planet since the very beginning of its spacecraft exploration. In addition to numerous experiments targeting atmospheric composition by remote sensing from the orbiter, such as Atmospheric Chemistry Suite (ACS) onboard ExoMars/ Trace Gas Orbiter (TGO) spacecraft [1], in situ measurements play an important role as well. In particular, atmospheric composition measurements onboard Mars rovers provide data to study complex chemical interactions at the atmosphere-subsurface interface [2], boundary layer meteorology [3], and even prebiotic synthesis and/or hypothetical biological activity on Mars [4]. All experiments listed above have been housed onboard mobile platforms and delivered information about atmosphere-surface interactions at different locations in the course of a rover's lifetime. Since the Viking landers activity (1976–1984) [5], no stationary platform has performed permanent long-term operation of both carbon dioxide and water vapor isotopologues. However, isotopic ratios in the atmosphere may substantially vary due to condensation/sublimation and adsorption desorption processes [6]. Thus, implications of laser spectroscopy technique for continuous monitoring isotopic ratios of main Martian volatiles remains an encouraging opportunity of Mars exploration.

　　　Following this task, the targeted molecules of the experiment are water vapor ($H_2O$), carbon dioxide ($CO_2$), and their isotopologues, in particular $HD^{16}O$, $H_2^{18}O$, $^{13}C^{16}O_2$, $^{16}O^{12}C^{17}O$ and $^{16}O^{12}C^{18}O$. The analysis of their spectral properties reveals that optimal intervals for spectroscopic measurements of all the molecules listed above are located in the vicinity of two wavelengths: 2.656 μm ($H_2O$, $HD^{16}O$ and $H_2^{18}O$) and 2.808 μm ($CO_2$, $^{13}C^{16}O_2$, $^{16}O^{12}C^{17}O$ and $^{16}O^{12}C^{18}O$). Synthetic transmission spectra of a near-surface layer of Martian atmosphere are shown in Figure 1. These spectra were calculated based on data published in [7–10], where the seasonal distribution of near-surface Martian atmosphere chemical composition is described. Figure 1 shows that absorption lines of all listed above molecules are available for measurement by one frequency sweep of the laser. However, main isotopic features, especially those of $HD^{16}O$, are weaker than one per cent for assumed measurement path length of 110 m. Therefore, relative noise level and systematic errors at the level of $10^{-5}$ or less are required for sensible measurements of isotopic ratios, which is a serious challenge to the instrument's design and data treatment techniques.

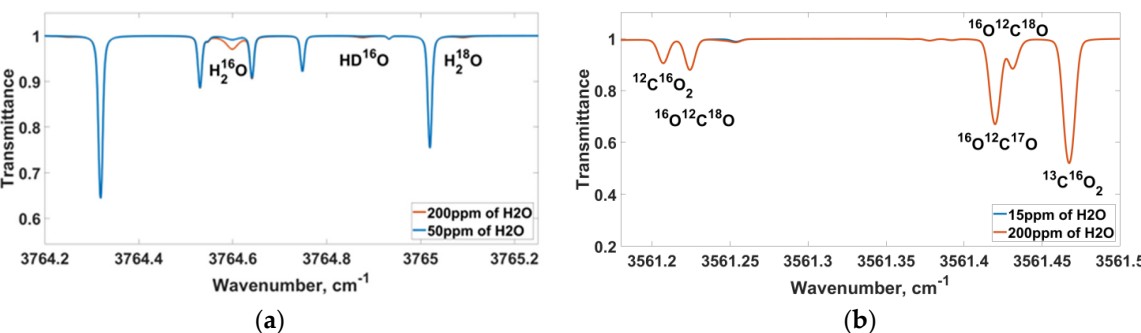

(a)　　　　　　　　　　　　　　　　　　　　　　　(b)

**Figure 1.** Synthetic transmission spectra of a near-surface layer of Martian atmosphere: (**a**) 2.656 μm at a pressure of 30 mbar; (**b**) 2.808 μm at a pressure of 5 mbar.

　　　In this paper, we describe the concept of the M-DLS instrument and present preliminary pre-flight calibration data in the context of future operation on the Mars' surface. The instrument's design is aimed at robust, easy-to-interpret and reliable in situ measurements, yet characterized with sensitivity fair enough for long-term isotopic monitoring of the volatiles.

## 2. Measurement Method and Experimental Setup Design

M-DLS employs off-axis integrated cavity output spectroscopy (OA ICOS) technique [11]. Two DFB lasers are used in the M-DLS for spectroscopic measurements of the Martian air probe in the vicinity 2.656 μm and 2.808 μm. The frequency of each laser is tuned continuously in the range of about $1\ \text{cm}^{-1}$ by current modulation. As there is a frequency sweep in the signal, the M-DLS measures fully resolved absorption line profiles of $CO_2$, $H_2O$ and theirs isotopologues with a spectral resolution of $\sim 0.002\ \text{cm}^{-1}$. The ICOS cell is exploited in the M-DLS for several reasons. Firstly, an ICOS cell is a compact construction, which allows reaching a long effective optical path needed to provide a high sensitivity of the M-DLS for detection of narrowly-spread isotopologues in the Martian atmosphere; secondly, the ICOS system has a construction resistant to the external impacts (strike, vibration), as compared with multipass optical systems.

It is essential to carry out the proper accumulation of the measured signal to reach a high SNR and measurement sensitivity of the M-DLS. An integration time of the majority diode laser spectrometers does not exceed the value of several seconds [12]. In our case, it is possible to accumulate a measured signal during periods of up to tens of minutes, since we exploit the ICOS cell as a reference channel, where absorption line peaks registered by the ICOS cell are used for synchronization of laser frequency tuning cycles. Such a method provides the level of the frequency stabilization less than $10^{-3}\ \text{cm}^{-1}$ and will be described further in a separate subsection.

The instrument functionality could be divided into three main systems, which will be described in details below:

- Martian air sample preparation system;
- Optical measurements system;
- Control electronics system.

The natural form of the M-DLS is presented in Figure 2a,b. The M-DLS operates in tandem with another instrument of the landing platform—MGAP (Martian gas analytical package), in particular, they have a joint air sample preparation system. This system contains two external interfaces—one for taking Martian air probe from the atmosphere into the spectrometer and one for sharing this probe with the MGAP. After air sample preparation procedure has been finished, the M-DLS makes spectroscopic measurements in the optical cell and transmits results to a landing platform computer BIP (a Russian acronym for the block of interfaces and memory).

The M-DLS has radiation-resistant unibody aluminum case with an oxide-fluoride coating. The whole weight of the device is about 3 kg with dimensions of $350 \times 188 \times 113.5$ mm (not including air intake device contribution to the length of the M-DLS). It is expected that the M-DLS is going to take measurements in the temperature range of −20 °C to 40 °C, which are provided by the landing platform.

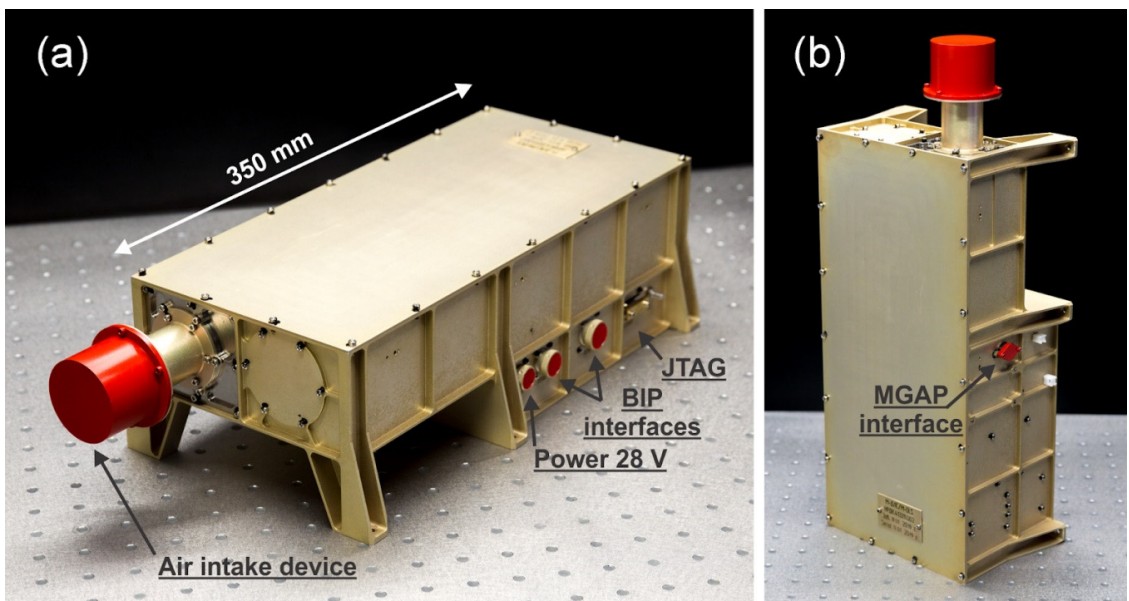

**Figure 2.** (**a**) View of the Martian multichannel diode laser spectrometer (M-DLS) with external interfaces power supply, BIP (a Russian acronym for the block of interfaces and memory), JTAG (joint test action group) and representative dimension; (**b**) the M-DLS view in its operating position on the landing platform with the MGAP (Martian gas analytical package) interface. Here the air intake device is held in transportation position, after landing it is going to be deployed 15 cm long.

## 2.1. Martian Air Sample Preparation

Preparation of the purified Martian air sample with certain values of its pressure and temperature is necessary for correct spectroscopic measurements. Therefore, the Martian air sample preparation system has a complex of devices to form a Martian air sample with necessary characteristics. The simplified air sample preparation system diagram is shown in Figure 3. This system includes an air intake device, gas line, four vacuum electromagnetic air sample regulation (ASR) valves, fine filters cassette, a vacuum pump, an ICOS cell volume, an electrical heater, pressure and temperature sensors and MGAP interface.

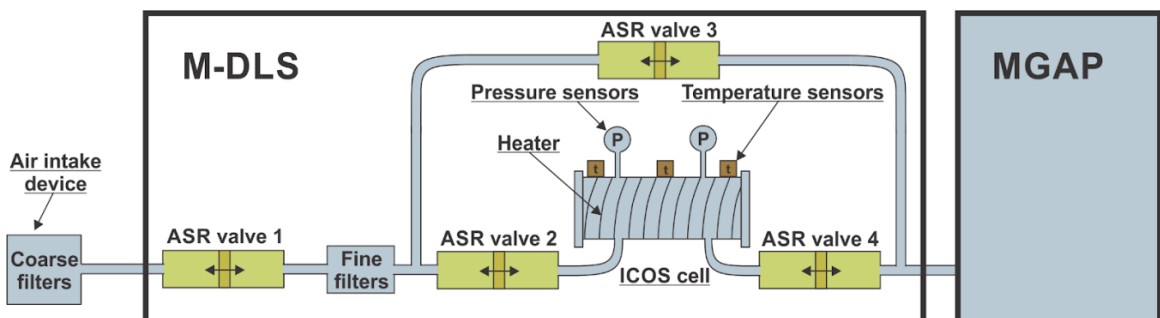

**Figure 3.** Principal Martian air sample preparation system diagram, joint for both the M-DLS and the MGAP (Martian gas analytical package) instruments. ASR valve is air sample regulation valve. The Martian air sample is purified by complex of coarse and fine filters. ASR valves 1, 2, 3, 4 and the MGAP pump control the Martian air sample update in the integrated cavity output spectroscopy (ICOS) cell. The ICOS cell contains multiple pressure and temperature sensors and is wrapped by nichrome wire (heater) for temperature maintenance.

The air intake device consists of a telescopic aluminum tube with coarse filters for dust protection. In the transportation position, this device is folded and locked, whereas in the working position,

the tube is deployed 15 cm long by a loaded spring and is fixed vertically up to 0.5 m above the landing platform. This allows us to take samples of an undisturbed air and reduce the inclusion of volatile components contaminated by the landing platform in the studied samples of air.

Electromagnetic valves, as well as a four vacuum pump, are designed by IKI RAS specifically to operate at low Martian ambient pressure. They are intended for the Martian air sample direction through the air sample preparation system to the ICOS cell before spectroscopic measurements. The pump, which is a part of the MGAP, is used for providing the M-DLS with Martian air sample at different pressures in the range of 5–30 mbar.

Each valve is represented by steel tube with a diameter of 3 mm, inside of which there are a magnetic core, a sealant and a limiter. The sealant and the limiter are located on opposite sides of the tube. The valve is closed when the core is on the sealant position, whereas when the core is on the limiter, the valve is open. Core movement is provided by a constant magnetic field, created by an inductor wound around the tube. Further explanation of valve switching can be found in the electronics subsection.

Fine filter cassette installed into the gas line is necessary for protecting vacuum electromagnetic pumps and high reflective ICOS mirrors against the dust. Metal powder filters are used as filtering elements. These filters are the 0.5 μm class solid-state disks with a diameter of 1.27 cm and a thickness of about 1 mm.

The resonator of the ICOS cell is a part of the air sample preparation system and is connected to it by two stainless steel tubes. Martian air sample is sent to the resonator with the volume of $V \approx 155.6$ cm$^3$ for spectroscopic measurements. During measurements, the temperature of this volume is maintained at 30 °C by the ICOS cell heater, while the actual values of Martian air sample temperature and pressure are measured by several kinds of sensors.

Two types of pressure sensors are used to measure the pressure level of the gas probe in the cuvette in different pressure ranges. The first type is the Heimann vacuum sensor HVS Vac 04 (HEIMANN Sensor GmbH, Dresden, Germany), which is a miniature Pirani-type sensor based on a heated resistor structure on a thin micromachined membrane. The second type is a miniature absolute piezoresistive pressure transducer XT-140 (Kulite Semicondactor Products, INC, Leonia, NJ, USA) based on standard miniature silicon diaphragm. The XT-140 is used for measuring the pressure higher than 30 mbar, while the HVS Vac 04 is exploited for low pressure measurements. These sensors were calibrated at different temperatures using high precision laboratory pressure sensors: ACM200 (Atovac, Yongin-si, Gyeonggi-do, Cheoin-gu, Korea) and Inficon DI200 (INFICON AG, Balzers, Liechtenstein).

There are also two types of sensors used for temperature measurements. The first one is a high precision semiconductor NTC MC65F103A temperature sensor (Amphenol Corporation, Wallingford, CT, USA). The second one is platinum temperature sensor 701-102BAB-B00 (Honeywell International Inc., Charlotte, NC, USA).

Thereby, one of each pressure sensors are installed inside the ICOS cell, as well as two MC65F103A and one 701-102BAB-B00 temperature sensors are installed outside the ICOS cell on its surface to precisely measure current values of pressure and temperature. In addition, there are pressure and temperature sensors inside the M-DLS to be aware of the instrument state during operation.

Thus, the Martian air preparation system allows us to form purified Martian air sample with the controlled values of its pressure and temperature, which is important for further spectroscopic measurements.

*2.2. Optical Measurements System*

The ICOS cell in the M-DLS carries two main functions: firstly, it is the main measurement instrument for spectroscopic research of Martian atmosphere; secondly, it is the reference channel for laser stabilization system to reach high spectral resolution and sensitivity. The sectional drawing of the ICOS cell is shown in Figure 4. The section is crossed along the main optical axis of the cavity throughout all mechanical, optical, and electronic components of the cuvette. As it could be seen in

Figure 4, the ICOS cell consists of following main components: a couple of DFB diode lasers (DL); complex of input and output optics, and the photodetector (PD) module. The ICOS cell represents a two-channel measurement system, as it contains two DFB lasers emitting on wavelengths of 2.656 μm and 2.808 μm with the optical power of 6–7 mW. The construction of the ICOS cell implies to employ one photodetector, therefore lasers operate in a sequential regime.

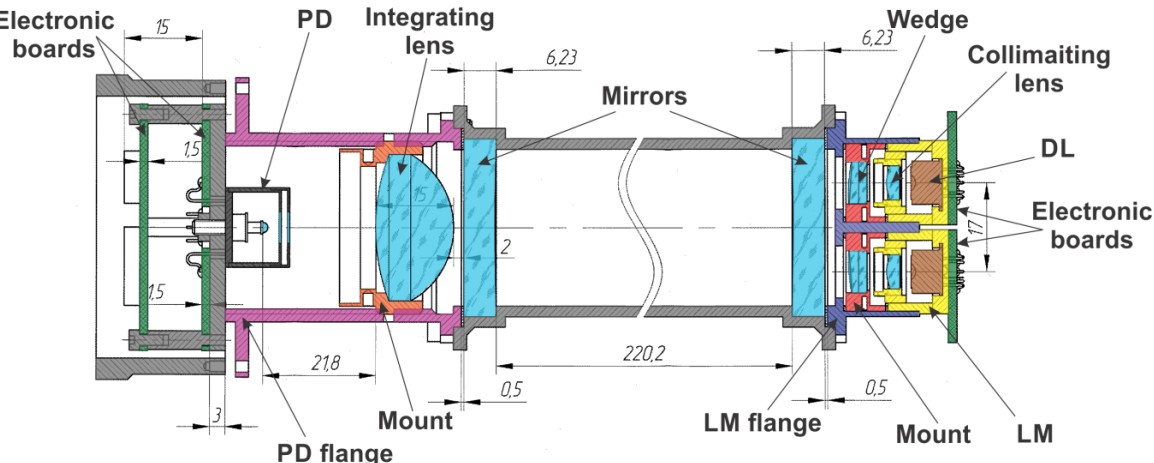

**Figure 4.** The sectional drawing of the ICOS cell, implemented in the M-DLS. Light blue parts show all optical components, including two collimating lenses, two optical wedges, two mirrors and the integrating lens. DLs (diode lasers) are brown, green elements are electronic boards, LM (laser module) constructions are yellow, optical wedge mounts are red and the LM flange is navy-blue. From the left cell side: PD (photodiode) is black, the integrating lens mount is orange and the PD flange is magenta. The ICOS cavity body and PD set are gray. (For interpretation of the references to color in this figure, the reader is referred to the web version of this article).

All mechanical components of the cuvette are made of aluminum with an oxide-fluoride coating for corrosion prevention, whereas mechanical parts which could interact with laser radiation have anodic oxidation coating, turning an aluminum surface into the porous surface with anti-corrosion and antireflection properties.

In more detail about laser modules, Nanoplus DLs (nanoplus Nanosystems and Technologies GmbH, Gerbrunn, Germany) in TO8 case with built-in thermoelectric cooler (TEC) and thermistors are exploited in the M-DLS. The DLs emit in quasi-continuous operation mode with the repetition frequency of spectral sweep about 70 Hz. The laser temperature tuning is used to set the DL to the targeted spectral range, while the current tuning provides continuous sweeping of the DL frequency in the range of 1 cm$^{-1}$. The DL is powered by pumping current pulse of the special form, which is shown in Figure 5. The first part of the current pulse is intended for laser crystal heating within 1.8 ms, where the current is higher than the threshold level to compensate for a nonlinear transient process of laser switching on. The second part is the frequency sweeping interval where the spectroscopic measurements are carried out. It is a sawtooth current pulse of 11.4 ms which consists of 1000 stepped points. The final part of the DL pulse is a time period with the duration of 1 ms when DL is turned off, this state of DL is used to register a dark current of PD and background radiation. Both DLs operate in the described mode, where each DL has its own unique set of pumping current parameters.

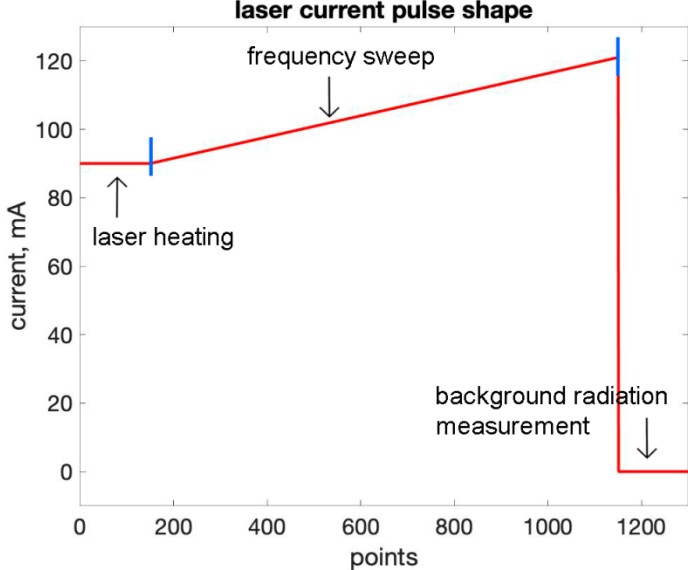

**Figure 5.** The schematic structure of the pumping current pulse, which consists of three stages: laser heating, frequency sweep, and background radiation measurement.

The manufacturer of DLs specified that the position of the crystal is not aligned with the center of TO8 case, thus it is mounted to the construction called laser module (LM) with the main function to form plane-parallel laser beam. LM consist of two elements: DL mounted on an aluminum plate with an electronic board for connection with atmospheric measurements unit (AMU); an aluminum mount with plano-convex collimating GeSbSe lens C036TME-D (Thorlabs Inc., Newton, NJ, USA) with diameter of 9.24 mm and working distance of 2.67 mm. LM is highlighted in yellow in Figure 4. The distance between the collimating lens in its mount to DL could be regulated by threaded connection with LM. The aluminum plate with installed DL allows aligning the optical axis of the collimating lens and the optical axis of DL because they are not centered by default.

After all tunings, we get a plane-parallel laser beam formed by LM, which goes to an optical quartz wedge with filtering dichroic coating. It is installed to a cylindrical aluminum mount (highlighted out by in red in Figure 4). The wedge has two plane sides: the input side is perpendicular to incident laser beam, while on the output side the plane is angled at 6.25° for directing the laser beam to the certain point on a mirror of the resonator. Particularly, by rotation of the wedge a signal pattern shape on resonator mirrors could be controlled. The signal pattern is a picture on the mirror formed by laser radiation, which consists of closely set spots where laser beam incidents and reflects, making up circle or ellipse. The quality of the signal passed through the ICOS cell depends on the signal pattern shape; a detailed analysis of it will be described and discussed elsewhere.

Two DLs together with a couple of wedges in their mounts are installed to the construction called LM flange (marked by navy-blue in Figure 4), which fixes an optical wedges' position and hermetically presses the input mirror to the body of the ICOS cell.

Moving on to the main part of the system, the cavity of the ICOS cell is a pair of spherical flat-concave quartz mirrors with a curvature radius of 500 mm located at a distance of 220 mm from each other. This distance is a little shorter than mirror's focal length of 250 mm. It is a typical configuration for the OA-ICOS, because this mirrors spacing, together with off-axis laser input to the cavity, allows to simultaneously pump many transverse modes to produce a dense cavity mode spectrum. Each mirror has an anti-reflection coating on the flat side with a reflection coefficient $R^* = 0.2\%$, and high reflection coating on the concave side with datasheet coefficients of $R_{2.6\ \mu m} = 99.8 \pm 0.1\%$ and $R_{2.8\ \mu m} = 99.6 \pm 0.1\%$ for 2.656 μm and 2.808 μm, respectively. The signal passed through the output mirror is collected and focused on the PD by a plano-convex quartz

integrating lens with a diameter of 28 mm and working distance of 21.8 mm. The integrating lens is installed into the aluminum mount (marked by orange in Figure 4), to set the distance to PD.

The HgCdTe photodiode PVI-3TE-3 (VIGO System S.A., Ożarów Mazowiecki, Poland) in TO8 case with built-in three-stage TEC, thermistor and input lens is exploited in the M-DLS for signal detection. The PD is mounted on the aluminum plate for heat dissipation and attached to two electronic boards for signal amplification and connection to the AMU. PD set with the integrating lens is installed into the construction called PD flange (marked by magenta in Figure 4), which fixes the lens position and hermetically presses the output mirror to the body of the resonator. All the optical components except the collimating lens were manufactured by OptoSigma Corporation, Santa Ana, CA, USA

Because of high reflectivity the input mirror transmits weak laser signal to the cavity, with a power about several thousandth from incident laser beam power. Passed signal, which was injected to the resonator by the optical wedge with a small angle to the optical axis of the cuvette, repeatedly reflects between mirrors providing a long effective optical path and forms a signal pattern on mirrors. The output mirror also transmits weak laser signal from the resonator, with power about several thousandths from incident signal power, thus the power of the cavity output signal is a million times weaker than the DL power. The signal from the output of the resonator is collected by the integrating lens and it effectively contains the information about every pass through the resonator. Hence, in the ICOS cell, we continuously register integral signal, accumulating information about each effective optical pathlength and, consequently, about each respective quantity of absorption in the investigated air sample. This integral signal is proportional to the average value of effective optical pathlength, which is calculated for a definite wavelength by the following equation:

$$L_{eff} = \frac{L}{(1 - R + \alpha L)} \tag{1}$$

where $R$—mirror's reflection coefficient, $L$—geometric length of the resonator, $\alpha$—per-pass intracavity absorption. In case of an empty cavity ($\alpha L = 0$) the gain factor for effective optical pathlength is defined by geometric parameters of the cavity. As the distance between mirrors is 220 mm then, according to the Equation (1), the effective optical pathlength is 110 m and 55 m for 2.656 µm and 2.808 µm respectively. Absorption medium in the cavity decreases the effective optical pathlength and if the per pass intensity of absorption is much less than cavity transmission ($\alpha L << (1 - R)$), ICOS absorption spectra can be described by the Beer–Lambert law. Otherwise, when these parameters are compatible ($\alpha L \approx (1 - R)$), the absorption line contour becomes wider with the absorption line magnitude depressed relative to the continuum level, the Beer–Lambert law doesn't work and more complicated approach is required.

The DL frequency scan rate is also an important parameter, which should be described before the spectrum analysis. According to the principle of operation, the ICOS is based on measurements of the steady-state power from the cavity, so establishing this state takes time $\tau$, which is defined by the cavity parameters and is called a cavity time constant:

$$\tau = \frac{L}{c(1 - R + \alpha L)} \tag{2}$$

Actually, $\tau$ is an average time interval within the light is captured by the optical cavity. If the time of DL frequency scanning over full width at half maximum (FWHM) of the spectral line is compatible with the cavity time constant ($\tau_{FWHM} \approx \tau$), it leads to a significant skewing of the line profile. In our case the DL scans the spectral range of 1 cm$^{-1}$ within 11.4 ms, thus the condition $\tau_{FWHM} >> \tau$ is met and the M-DLS measures cavity transmittance spectra without line profile distortion.

## 2.3. Control Electronics System

The electrical part of the device is represented by several boards, connections and separate modules shown in Figure 6. Two main boards named the atmospheric measurements unit (AMU) and

the service system unit (SSU) are responsible for handling an experiment and basic functionality of the M-DLS.

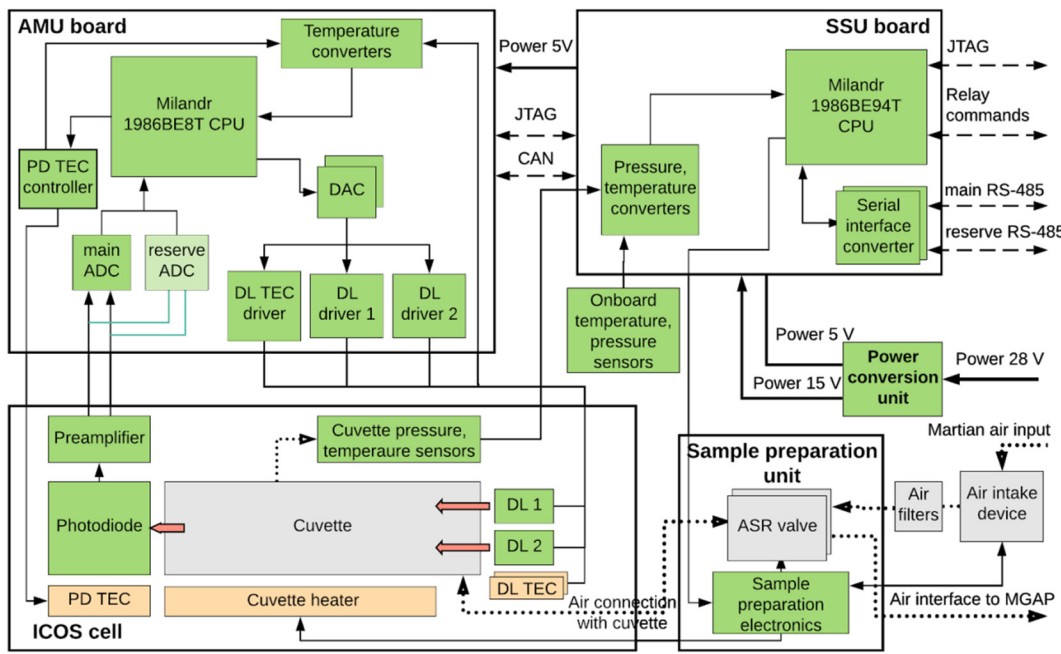

**Figure 6.** Block diagram of electric circuits of the M-DLS. There are four main blocks: the atmospheric measurements unit (AMU) for experiment handling; the service system unit (SSU) for auxiliary systems operation and data transfer to the landing platform; the air sample preparation unit and the ICOS cell for spectroscopic measurements.

The AMU performs driving of diode lasers (DLs) and acquisition of photodiode (PD) signal, thus carrying the main role in experiment conduction. A laser driver circuit is based on a single transistor scheme and provides currents of up to 200 mA. One 16-bit aerospace DAC is used for the current setting with the sampling frequency of 87 kHz. Every DL is equipped with the TEC for the temperature setting, which is controlled by similar DAC with the sampling frequency of 70 Hz. For the temperature control, there are thermistors inside and outside the DL case. A simple differential amplifier is used for the temperature conversion within 20 to 45 °C with a multichannel 12-bit analog-to-digital converter (ADC) built in the microcontroller in oversampling mode. Temperature readings are then proceeded to the digital proportional-integral (PI) controller to maintain a selected constant laser temperature with an error less than 5 mK. As for the photodiode, its operation requires analog signal digitalization and temperature control by the AMU. Preamplified PD signal is digitized by a 16-bit industrial ADC on the AMU with the sampling frequency coinciding with the DACs frequency of 87 kHz. For the PD to function in regime with the highest SNR and sensitivity it is necessary to cool it down to the temperatures around −31 °C using the inbuilt TEC. Since the temperature of PD is not obligated to be retained precisely, a simple circuit with an analog PI controller was selected. The core of the AMU is represented by the radiation and heat resistant Milandr 1986BE8T Cortex-M4 microcontroller ("PKK Milandr" joint-stock company, Zelenograd, Russia) running at 40 MHz with "write once read many" memory type. Its performance was enough for simple real-time operations with collected spectral data.

The SSU is the second main board in M-DLS, responsible for the M-DLS communication with the BIP, for relaying data from BIP to AMU, and for the operation of all auxiliary systems. Being responsible for the control of auxiliary systems, the SSU converts data from all pressure and temperature sensors. To get the pressure sensors readings a Wheatstone bridge arrangement was used, the output signal is fed to a differential amplifier and digitized by a 12-bit ADC. The SSU also controls the sample

preparation electronic board. The core of the board is a heat resistant Milandr 1986BE94T Cortex-M3 microcontroller running at 24 MHz. As for the serial interfaces, all input packets from BIP are received by SSU from full-duplex RS-485 with a second reserve line. These input packets are either intended for the SSU or relayed to the AMU by CAN bus between the SSU and AMU.

The ICOS cell, which is highlighted as separate section in Figure 6, from the electronics point of view consists of several components: the PD, two DLs, the cuvette heater, and temperature/pressure sensors. Photocurrent from the PD is converted into a voltage by a transimpedance amplifier (TIA). A bootstrapping amplifier is set along TIA to lower PD capacity and increase its bandwidth. Then the PD signal passes through a low-pass filter of 200 kHz with a variable gain of 1 or 2 and a single-to-differential converter at the end. Variable gain is implemented because the signal amplitude depends on the choice of DL channel, in addition, background temperature significantly affects the dark signal level due to high sensitivity of PD, decreasing available signal range. As for the cuvette heater, which consists of nichrome spiral thread, it is powered by the sample preparation electronics. The temperature of the cuvette is controlled by the SSU and is set about 30 °C. Thermal isolation of the ICOS cell allows heating the cuvette with the rate of 2.5 °C per minute.

The next block in Figure 6 with its electronics is a sample preparation unit. The main task of this board lies in the solenoid valves switching, which is implemented by four combined H bridges and a charge pump. With the current flow direction, the valve either closes or opens, wherein both positions are stable. The charge pump generates a necessary voltage for the operation of the valve and includes a cascade of capacitors with a total output capacity of 1880 μF. To switch each valve successfully a voltage pulse of 20 V with a duration of more than 10 ms is required. There is also an air intake device electronics connected to the sample preparation unit, which consist of resistors and two reed switches, responsible for a thread burning of the rod and its position control.

Finally, the power conversion unit provides the power for the whole device by converting 28 V to 15 V and 5 V. There are many other devices on the landing platform, which could add interference into power signal even with the separate power line for M-DLS. Therefore, one of the most important tasks of this electronic board is input power filtration.

### 2.4. Line Stabilization Algorithm

As it was stated earlier, long-time signal accumulation, which is necessary for high spectral resolution and measurement accuracy in the ICOS cell, is only possible with proper laser frequency stabilization. Thus, line stabilization mode is necessary if spectrum averaging is enabled.

Common laser temperature control is based on thermistor input, which is not precise enough and does not represent actual laser temperature, rather some intermediate inert value, therefore spectrum averaging in this mode leads to a line profile deformation, decreasing the accuracy of the model fitting. To solve this problem, we implemented a laser temperature controller based on its spectral information. The algorithm is aimed at spectral line center search in the photodiode output signal, shown in Figure 7a. The error between the actual and desired line center coordinates is then given to the PI controller for laser TEC control, leading to a spectral image fixation.

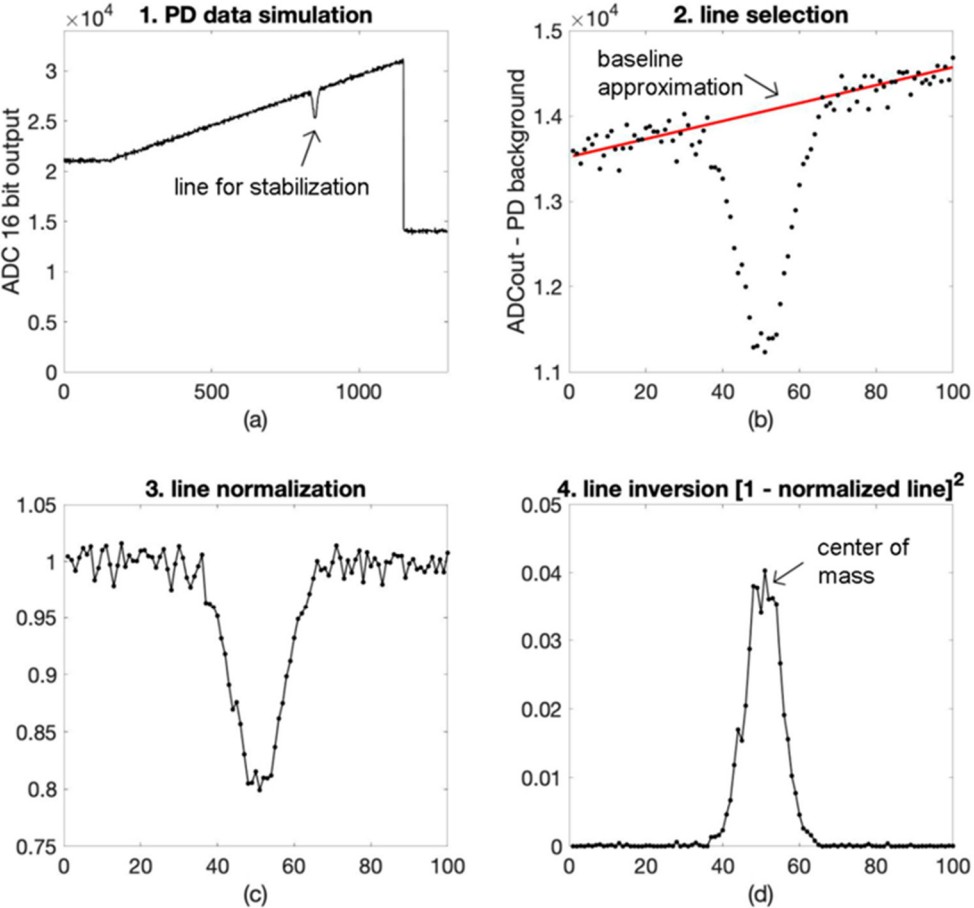

**Figure 7.** Real-time line center search algorithm visualization. (**a**) simulation of photodiode data for algorithm verification; (**b**) slice of original data with deducted background radiation; (**c**) line after normalization; (**d**) final result for the center of mass calculation.

Because of limited computing resources and a small number of points (less than 100) per line profile, a simplified algorithm was implemented. Firstly, a predefined part of the spectrum is sliced out and background radiation is deducted, as shown in Figure 7b. Secondly, a baseline is calculated using 15 edge points for later normalization, which could be seen in Figure 7c. After line normalization, the spectrum is deducted from 1 and squared to get a positive bell for the center of mass calculation, seen in Figure 7d. Using Formula (3), a line center is obtained and error between actual and desired centers enters the laser TEC PI controller.

$$Line\ center = \frac{\sum_i i \times Spectrum_i}{\sum_i Spectrum_i} \tag{3}$$

A simulation of a line center search algorithm gave the accuracy of 0.1–0.2 of the interval between consecutive measurements, whereas the real implementation occurred to 4-fold less accurate. Such precision drop is mainly caused by an optical interference on a spectrum and PI controller precision, nevertheless, providing a temperature-based controller is applied, we observe the 1–2 intervals accuracy with noticeable long-time spectral lines position drift. As for the rough conversion to a physical value, 1 spectral interval equals to 0.001 cm$^{-1}$. It is also interesting to note that the second-order least-squares method for line center approximation showed worse results than the center of mass algorithm in simulation.

## 2.5. M-DLS Measurement Cycle

After touchdown of the ExoMars-2022 landing platform on the surface of Mars and the rover egression, we prepare for further experiments by deploying the air intake device. Following successful deployment, the instrument is ready to operate. The principal scheme of the whole M-DLS and its internal layout is shown in Figure 8a,b. The experiment begins from the preparation of Martian air sample. This procedure is conducted by both the M-DLS and MGAP instruments. The Martian air preparation system can operate in two regimes: the first is a preparation of air sample at Martian ambient pressure; the second is at higher pressures up to 30 mbar. There are sufficient differences between these modes, therefore, they will be described separately. All operations during the Martian air sample preparation procedure are provided by the SSU under commands from the BIP.

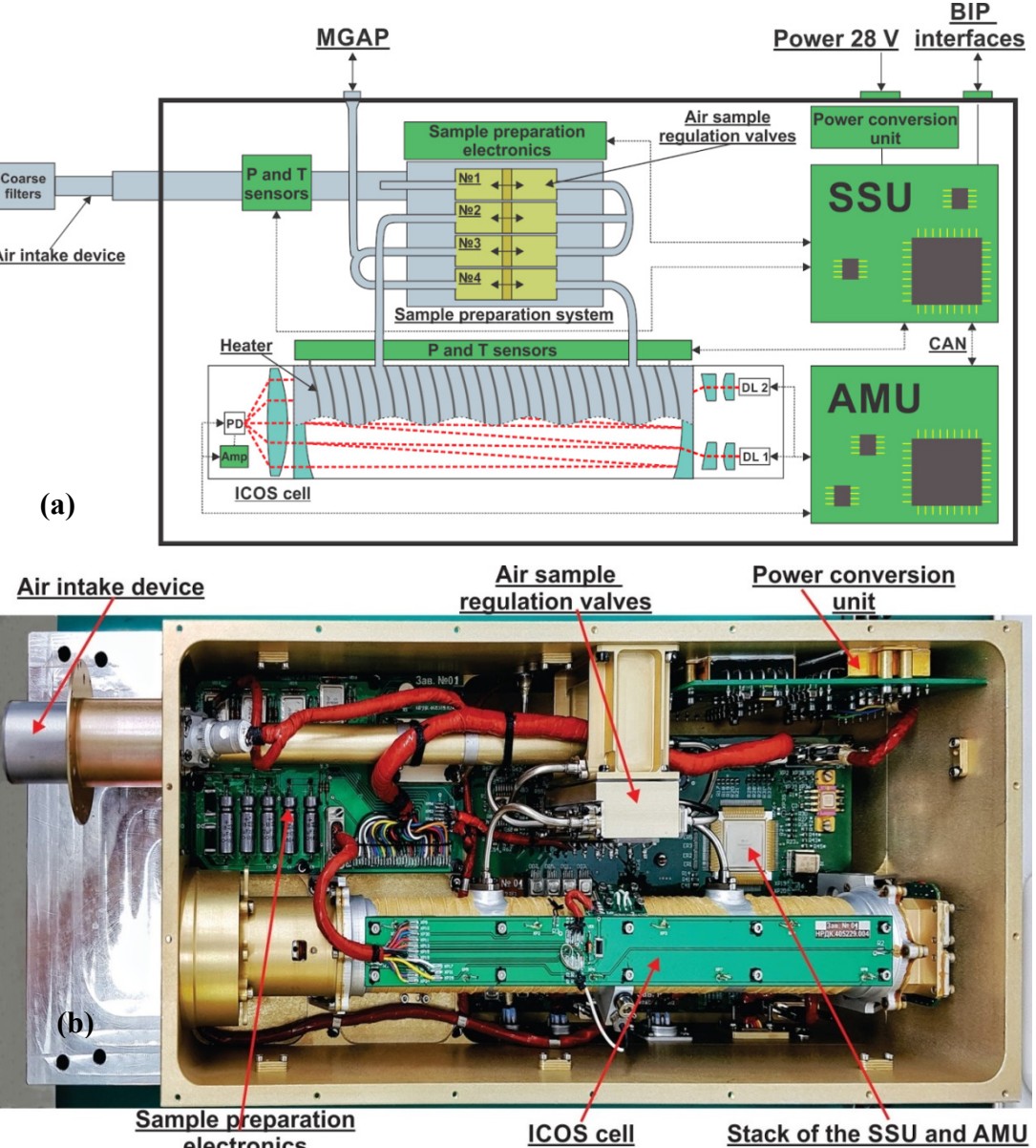

**Figure 8.** Principle scheme (**a**) and internal layout (**b**) of the M-DLS. SSU is service system unit, AMU is atmospheric measurements unit.

The procedure of the Martian air sample preparation begins with the pressure and temperature sensors interrogation to evaluate the current condition of the instrument. Then, the SSU, with the

help of sample preparation electronics, sets the ASR valves position to a configuration which allows pumping out the ICOS resonator volume under the $10^{-4}$ mbar by the MGAP vacuum pump. The next step is to configure valves to provide a Martian air with the path to the ICOS cell. Then the MGAP turns off the pump. After that, the M-DLS configures valves to block the cavity volume from the outside atmosphere and to equalize pressure level in the whole gas system. Thereafter, the heater is turned on to reach a predetermined temperature. The final step of the air sample preparation procedure is pressure and temperature sensor data acquisition.

In case we understand the M-DLS sensitivity is not satisfied to our measurement purpose due to lack of information about $H_2O$ and $CO_2$ isotopologues content near the surface, there is an opportunity to increase the instrument sensitivity by increasing the pressure of the Martian air sample up to 30 mbar. The main difference of operation in this regime is that after pumping out the cavity volume, the M-DLS closes all valves, while the MGAP turns off the pump and connects the ballast tank with Martian air at a high pressure of 930 mbar to the M-DLS gas system. Then the M-DLS configures valves to fill the cavity with Martian air from the tank.

As soon as Martian air sample preparation is over and experiment setup data packet from BIP is received by the SSU, it is relayed to the AMU for spectroscopic measurements initialization. After that, peripherals are set and AMU enters the preparation stage for 3 min, where PD cooling is enabled, selected DL goes to the predefined temperature and laser current modulation is turned on to enter the stable thermal regime. The DL temperature is handled by the PI controller based on thermistor data input. After 3 min a line stabilization is linked up to control DL temperature precisely throughout the experiment. The AMU enters a data acquisition stage after a minute of line stabilization, where spectra are collected for up to 15 s each with averaging of up to 700 times. Following the acquisition of 7 spectra, the AMU enters a hibernation mode at the 6th minute of the experiment, waiting for packets request from SSU.

It is worth to notice that the air preparation procedure is highly dependent from environmental conditions, preparation time can take up to 90 min, including only two invariable parameters: 6 min for one configuration of ASR valves and 1.5 s for the sensor interrogation. In case of malfunction of air sample preparation system, there is a scenario where M-DLS functions in the mode with only diffusion responsible for air sample update.

Experiment data volume with the 7 measured spectra for a single laser from the AMU is around 14 Kbyte (7 spectra per 2032 byte). The SSU sensor data volume also is 2032 byte. If the M-DLS operates once per sol within several hours, the output data will contain up to 28 spectrum packets and numerous data packets from the SSU. The total volume of data received within one sol will not exceed 500 Kb. Power consumption does not exceed 15 W, 12 W in heating mode, and 10 W in spectra acquisition mode.

Important technical details of the M-DLS are presented in Table 1.

**Table 1.** Technical parameters of the M-DLS.

| Size, mm | Weight, kg | Power Consumption, W | Spectral Range, μm | Targeted Molecules | SNR | Spectral Resolution, cm$^{-1}$ |
|---|---|---|---|---|---|---|
| $350 \times 188 \times 113.5$ [1] | 3 | 15 [2] | 2.656 | $H_2O$, $HD^{16}O$, $H_2^{18}O$ $CO_2$, $^{13}C^{16}O_2$, $^{16}O^{12}C^{17}O$, $^{16}O^{12}C^{18}O$ | 7500 | 0.002 |
| | | | 2.808 | | 3000 | 0.001 |

[1] Not including air intake device contribution to the length of the M-DLS. [2] Maximal peak power.

## 3. Measurement Results

Measurement capabilities of the M-DLS were verified on the non-referenced gas mixture of the carbon dioxide and water vapor with proportions of 90% and 10% for $CO_2$ and $H_2O$ respectively. Actually, for spectroscopic measurements, the optical cell was filled with pure $CO_2$, but approximately 0.5 mbar of $H_2O$ remained in the cell due to desorption abilities of inner surfaces of the cell. The M-DLS

was placed in a vacuum gas chamber filled with $CO_2$ under the pressure of 5 mbar for spectroscopic measurements. As the measurements were carried out in the transportation position of the M-DLS where the air intake device is folded, gas samples for the measurements were injected to the optical cell separately from the main volume of the gas chamber by the MGAP interface. The sequence of the experiment was following: at first, gas sample was injected to the optical cell by the long steel tube (length 1 m, diameter 3 mm) under the control of the precise pressure sensor Inficon DI200; then there was a period of 10 min to reach homogeneity in the optical cell; after that control packets were sent to the SSU to acquire data from pressure and temperature sensors; if the pressure of the gas sample in the optical cell satisfied expected value, experiment setup packets were sent to the AMU for spectroscopic measurements in the range of 2.656 μm, after 6 min of experiment AMU returned 7 spectra, each of which consists 300 averaged spectra; then the SSU additionally conducted pressure and temperature sensor data pulling, and the AMU repeated the spectroscopic experiment for the second spectral range of 2.808 μm. This experiment was repeated for several gas sample pressure levels, but in this paper, we present data registered at expected ambient Martian air pressure level of 5 mbar.

Spectroscopic measurements were carried out after all preflight tests, in particular, after tests on vibration test bench, which confirmed the ability of the instruments optics to withstand space travelling. One of seven acquired spectra for each laser, which consists of 300 averaged measurements, is represented in Figure 9a,b for 2.656 μm and 2.808 μm respectively (blue curves). The second order polynomial was used for the baseline approximation (red curves). The linearization of the frequency domain of registered spectra was based on the linear interpolation of frequency intervals between strong absorption line peak positions. In our case, exploiting of the temporary Fabry–Perrot etalon for such purpose is impossible because of inconstant frequency nonlinearity profile of each laser, we observed nonlinearity profile's form change after thermal–vacuum tests and even after hourly work with the instrument. Presented spectra were registered under the pressure level of 4.75 ± 0.008 mbar and 4.89 ± 0.008 mbar for 2.656 μm and 2.808 μm, respectively. The temperature of the optical cell was measured by three sensors. Averaged value of these sensors was set as gas sample temperature, which was 303 ± 0.002 K for both spectral ranges.

Normalized and linearized experimental spectra (blue curves) with the synthetic spectra (red curves), as well as their residuals are shown in Figure 9c,d for 2.656 μm and 2.808 μm respectively. In contrast to raw spectra in Figure 9a,b, normalized spectra are a result of averaging of seven AMU measurements per single measurement cycle. Parameters of spectral lines for the synthetic spectra were taken from HITRAN 2016 [13], Voight profile was exploited as line contour shape. As spectral ranges were chosen in such a way to include each spectral line of interest in separate, not an overlapped position, fitting of the experimental spectra is carried out by linear regression method. All spectral lines inside the observed spectral range were included in synthetic spectra. Self-broadening parameters were used for $CO_2$ lines, whereas air-broadening parameters were used for $H_2O$ due to lack of data on the pressure broadening of $H_2O$ in $CO_2$ in our spectral regions.

Preliminary retrieved data of isotopologue abundance ratio from measured spectra in comparison with HITRAN isotopologue abundance are presented in Table 2. Isotopologue abundance ratio was calculated based on the following equation (on the example of $^{18}O$ contained in water vapor):

$$^{18}R_{(H_2O)} = \frac{N(^{18}O)}{N(^{16}O)} \tag{4}$$

where $N(^{18}O)$—number of water vapor molecules with $^{18}O$, $N(^{16}O)$—number of water vapor molecules with $^{16}O$. Comparison of the retrieved isotopologue abundance ratio with the ratio from *HITRAN* was calculated by the equation (on the example of $^{18}O$ contained in water vapor):

$$\delta^{18}O_{(H_2O)} = \left( \frac{^{18}R_{(H_2O)}}{^{18}R_{(H_2O)_{HITRAN}}} - 1 \right) \times 1000 \tag{5}$$

where $^{18}R_{(H_2O)\_HITRAN}$—oxygen isotopologue abundance ratio of the *HITRAN*. As for the retrieval precisions of the isotopologue abundance ratio, it was calculated based on a variation of parameters included to the model and are 0.87% and 3.35% for 2.656 μm and 2.808 μm respectively.

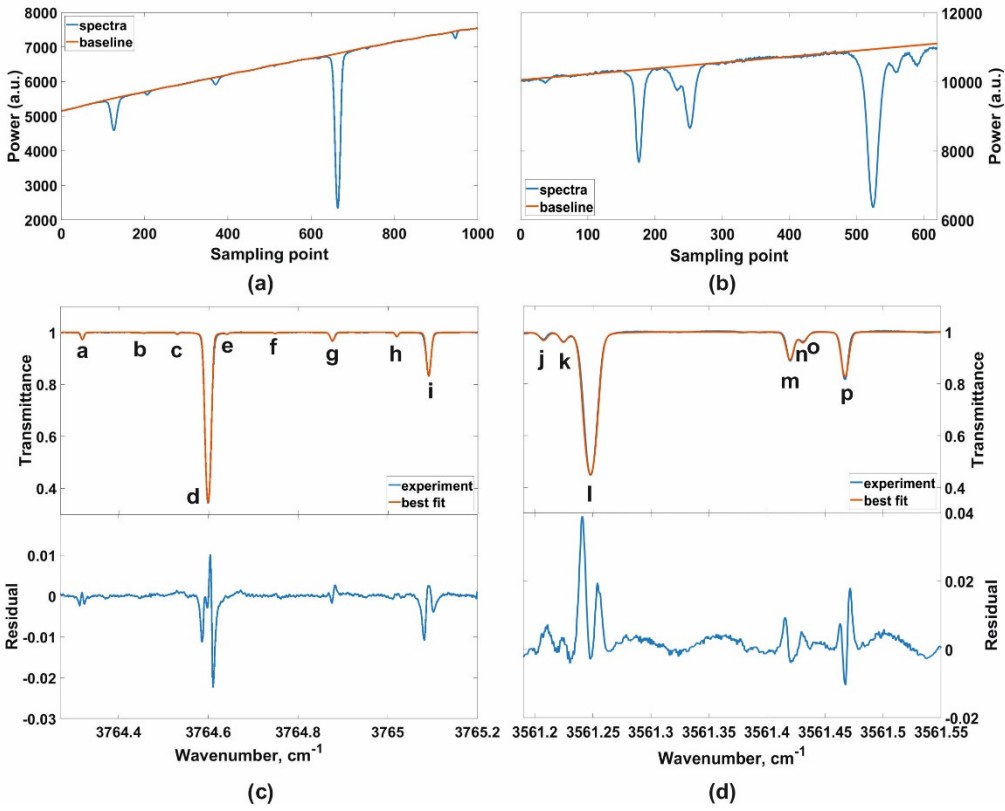

**Figure 9.** Experimental data of the M-DLS. (**a**,**b**) raw spectra (blue curves) and approximated baselines (red curves) for 2.656 μm and 2.808 μm respectively. (**c**,**d**) normalized transmittance spectra (blue curves) with synthetic spectra (red curves) and residual at separate graphs (blue curves) for 2.656 μm and 2.808 μm, respectively. Line identifiers from "a" to "p" are deciphered in the discussion section. (For interpretation of the references to color in this figure, the reader is referred to the web version of this article).

**Table 2.** Comparison of the isotopologue abundance ratio and retrieval precision of the M-DLS.

| Wavelength, μm | Abundance Ratio, ‰ | | Retrieval Precision, % |
|---|---|---|---|
| 2.656 | $\delta^{18}O_{(H_2O)}$ | 28.4 | 0.87 |
| | $\delta^{2}H_{(H_2O)}$ | 127.1 | |
| 2.808 | $\delta^{13}C_{(CO_2)}$ | 170.6 | 3.35 |
| | $\delta^{18}O_{(CO_2)}$ | 7.8 | |
| | $\delta^{17}O_{(CO_2)}$ | 87.7 | |

## 4. Discussion

The main purpose of this paper is to present the M-DLS and describe its structure and modes of operations, therefore, the data shown in Table 2 should be treated with reservations for several reasons: firstly, assembling, tuning, and testing of the M-DLS were carried out in a limited amount of time, so the tests on an etalon gas mixture were not yet conducted, consequently, for now, there is a lack of demonstration of the accuracy of the M-DLS; secondly, synthetic spectra are based on HITRAN database parameters, which should be validated, and Voight profile is used for modeling, which does not properly represent real conditions; and finally, the water vapor spectral lines, which are presented

in the measured spectra, are conditioned by absorbed water vapor, i.e., water vapor, which appear in the optical cell from the porous structure of its internal surfaces during pumping out.

For further discussion, it is reasonable to demonstrate the HITRAN list of spectral lines, which contribute the most to the absorption and are used for isotopologue abundance ratio retrieval. Isotopologue abbreviation and central frequency of its transition for both spectral ranges are presented in Table 3.

**Table 3.** Isotopologue abbreviations and central frequency of its transition for both spectral ranges of the M-DLS. Identifiers are according to Figure 9.

| 2.656 µm | | | 2.808 µm | | |
|---|---|---|---|---|---|
| **Identifier** | **Isotopologue** | **ν, cm$^{-1}$** | **Identifier** | **Isotopologue** | **ν, cm$^{-1}$** |
| a | $^{12}C^{16}O_2$ | 3764.3188 | j | $^{12}C^{16}O_2$ | 3561.2072 |
| b | $^{12}C^{16}O_2$ | 3764.5304 | k | $^{16}O^{12}C^{18}O$ | 3561.2243 |
| c | $H_2^{16}O$ | 3764.599130 | l | $H_2^{16}O$ | 3561.247700 |
| d | $^{12}C^{16}O_2$ | 3764.6409 | m | $^{16}O^{12}C^{17}O$ | 3561.4199 |
| e | $^{12}C^{16}O_2$ | 3764.7484 | n | $H_2^{16}O$ | 3561.42841 |
| f | $HD^{16}O$ | 3764.87629 | o | $^{16}O^{12}C^{18}O$ | 3561.4312 |
| g | $^{12}C^{16}O_2$ | 3765.0198 | p | $^{13}C^{16}O_2$ | 3561.467 |
| h | $H_2^{18}O$ | 3765.090810 | - | - | - |
| i | $^{12}C^{16}O_2$ | 3765.2642 | - | - | - |

The characteristic shape of water vapor lines residual in both spectral ranges of the M-DLS in Figure 9c,d indicates that exploiting of air-broadening parameters at half maximum, listed in the HITRAN and application of Voight profile for the model, at this stage does not fully satisfy our expectations for modeling of the measured spectra.

We keep in mind that the M-DLS could influence on the line profiles, but the problem is that the M-DLS is installed on the landing platform. Despite on transfer of the ExoMars mission launch to 2022, deinstallation of the M-DLS and additional manipulation with it are restricted. Therefore, the instrument influences on line profiles will be investigated during measurements of the Martian air sample by changing the averaging number of a single measurement.

During the mission, the ICOS cell mirrors are expected to be gradually contaminated by the Martian atmospheric dust, with their reflectivity being degraded and the cell's effective optical path shrunken. However, the main scientific goal of the experiment is isotopologues ratios measurement, rather than absolute abundances. As we measure all spectral lines of interest, including those of main isotope and minor ones, within a single laser frequency scan, a change of the effective optical pass is not a problem. Moreover, the main isotopologue of $CO_2$ is quite a stable component of the Martian atmosphere, and its lines present in both spectral ranges, this fact, along with the accurate measurement of pressure with the precision of 0.2%, allows us to calibrate the measurements with the required accuracy.

Brief spectroscopic analysis of measured data presented in the previous section is an approximate intermediate step of a complex process of data analysis. There are numerous steps for further improvements of data processing. First of all, separate precise measurements of line parameters from Table 3 should be conducted for clarification. Secondly, Fourier filtration of the experimental spectra could be applied for low-frequency interference deliverance that is well noticed in the residual of 2.808 µm range (Figure 9d). More complex line profile models for synthetic spectra, for example, Galatory profile [14], could also increase the fitting quality of the model. Furthermore, and finally, we will conduct analysis of apparatus distortion of the line profiles. Only after that it will be possible to give a proper estimation about the accuracy characteristics of the M-DLS. Part of this work is already underway and will be published elsewhere.

For future development of the method, we plan to consider ICOS cells with longer effective optical path and optimized raytracing inside the cell to damp the interference. These measures may

potentially increase the instrument's sensitivity at least by one order of magnitude. In addition, we plan to employ ICOS cell as a reference channel for spaceborne NIR multichannel heterodyne laser spectroradiometer [15], which will be described elsewhere.

## 5. Patents

The M-DLS concept has been patented as "M-dls Martian Multichannel Diode-Laser Spectrometer" RU2730405C1 in 2020.

**Author Contributions:** Conceptualization and methodology, A.K., V.S., G.D., M.G.-D., M.S., T.K. and Y.L.; software, I.G. (Iskander Gazizov), V.K., A.V., V.B.; validation, S.Z., I.G. (Iskander Gazizov), V.K. and V.M.; formal analysis, S.Z., I.G. (Iskander Gazizov) and V.K.; investigation, S.G., I.G. (Iskander Gazizov), V.K., V.M., I.G. (Ilya Golovin), S.M., A.N. and O.R.; data curation, S.Z., I.G. (Iskander Gazizov) and V.K.; writing—original draft preparation, S.Z., I.G. (Iskander Gazizov) and A.R.; writing—review and editing, V.K., M.S. and V.M.; visualization, S.Z., I.G. (Iskander Gazizov) and E.T.; supervision, I.V., A.R. and O.K.; project administration, V.K. and I.V. All authors have read and agreed to the published version of the manuscript.

**Funding:** This work was supported by the Ministry of High Education and Science of Russian Federation, for subsidy 0028-2019-0016 to IKI RAS and by the Russian Foundation for Fundamental Investigations (18-29-24204 (Maxim Spiridonov, Iskander Gazizov, Vyacheslav Mesherinov), 19-32-90276 (Sergei Zenevich)). Industrial works for the ExoMars-2020 mission were supported by the ROSCOSMOS.

**Acknowledgments:** Authors are grateful for current Roscosmos contract support of the ExoMars mission industrial work and for the CNES support of the M-DLS laboratory research work.

**Conflicts of Interest:** The authors declare no conflict of interest.

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
