# Peer review of "Martian Multichannel Diode Laser Spectrometer (M-DLS) for In-Situ Atmospheric Composition Measurements on Mars Onboard ExoMars-2022 Landing Platform"

_applsci, doi:10.3390/app10248805_

Round 1
Reviewer 1 Report
The paper describes a laser spectrometer operating in the mid-IR spal range, at 2656 nm and 2808 nm, for long-term monitoring of isotopic ratios in main Martian volatiles - carbon dioxide and water vapor – in the near-surface atmosphere. The detailed description of the spectrometer is presented.
The questions and suggestions are as follows.
- Figure 4 demonstrates a schematic of optical measurement system. The optical cavity on the Figure 4 has two plane mirrors. However, according to description on the page 7 the cavity is formed by two plane-concave mirrors with near to confocal configuration. Thus, the schematic of the cavity on the figure 4 doesn’t match its description.
- The power of the laser diodes at 2656 nm and 2808 nm should be done. It is important for an estimation of the intracavity optical power in the presence of strong optical losses on the input mirrors.
- A sensitivity of the spectrometer to angle detuning of the diode laser beam (with respect to the cavity axis) should be estimated. Such angular misalignment can occur when overloading during Mars landing.

Author Response
We agree with the most recommendations and thank the reviewers who helped us to improve the manuscript. We present hereafter the corrected sentences with modified text marked by yellow.
The paper describes a laser spectrometer operating in the mid-IR spal range, at 2656 nm and 2808 nm, for long-term monitoring of isotopic ratios in main Martian volatiles - carbon dioxide and water vapor – in the near-surface atmosphere. The detailed description of the spectrometer is presented.
The questions and suggestions are as follows.
- Figure 4 demonstrates a schematic of optical measurement system. The optical cavity on the Figure 4 has two plane mirrors. However, according to description on the page 7 the cavity is formed by two plane-concave mirrors with near to confocal configuration. Thus, the schematic of the cavity on the figure 4 doesn’t match its description.
Indeed, in Figure 4 we presented the sketch of the cell in realistic proportions. One can notice a slite concave surface on the inner sides of mirrors, such a small bending is caused by 500 mm of the curvative radius. For the visualization purpose, we additionally plot the scheme of the optical cell structure in Figure 8a, where we highlighted the curvatives of the inner sides of mirrors.
- The power of the laser diodes at 2656 nm and 2808 nm should be done. It is important for an estimation of the intracavity optical power in the presence of strong optical losses on the input mirrors.
We added information about LD power:
Lines 181-183: The ICOS cell represents a two-channel measurement system, as it contains two DFB lasers emitting on wavelengths of 2.656 µm and 2.808 µm with the optical power of 6-7 mW.
- A sensitivity of the spectrometer to angle detuning of the diode laser beam (with respect to the cavity axis) should be estimated. Such angular misalignment can occur when overloading during Mars landing.
Yes, this is an important point. We evaluated such mechanical misalignments, which leads to significant deformations of the signal pattern on the mirrors and as the result bring sufficient decreasing of spectrometer sensitivity. But we are going to leave it as a subject of a separate study, which, along with a detailed spectroscopic analysis, will be published elsewhere. We noticed it on page 7 lines 225-227.

Reviewer 2 Report
This paper presents a multichannel diode laser spectrometer for In-situ atmospheric measurements on Mars onboard ExoMars-2022 landing platform. The spectrometer is designed with ICOS technique using two DFB lasers near 2.6 um and 2.8 um for the sensitive detection of H2O and CO2 isotopes, respectively. The authors gave very detailed discussion of the instrument development including air sample preparation, optical measurements system, control electronics, laser stabilization, as well as the representative results and relevant discussions. The paper is well written and organized and should be of great interest to the community. The reviewer recommends the publication of this paper with a minor revision.
- Fig. 2 and other relevant figures: please add the full names in the caption for those abbreviations in the figure.
- The ICOS performance will be significantly affected by the reflectivity of the mirrors of the gas cell. The effective absorption path length is determined by the reflectivity. It is possible that the mirror can be contaminated during the 1-year test. Can the authors comment on this issue? Then how to calibrate the instrument?
- The instrument has a power consumption of 15 W. Can the authors comment about the power supply? Battery integrated with solar panel?
- It will be an impressive work if the proposed instrument can be used in the future space exploration.
Author Response
We agree with the most recommendations and thank the reviewers who helped us to improve the manuscript. We present hereafter the corrected sentences with modified text marked by yellow.
This paper presents a multichannel diode laser spectrometer for In-situ atmospheric measurements on Mars onboard ExoMars-2022 landing platform. The spectrometer is designed with ICOS technique using two DFB lasers near 2.6 um and 2.8 um for the sensitive detection of H2O and CO2 isotopes, respectively. The authors gave very detailed discussion of the instrument development including air sample preparation, optical measurements system, control electronics, laser stabilization, as well as the representative results and relevant discussions. The paper is well written and organized and should be of great interest to the community. The reviewer recommends the publication of this paper with a minor revision.
- 2 and other relevant figures: please add the full names in the caption for those abbreviations in the figure.
Abbreviations were described:
Lines 106-109: Figure 2. (a) View of the M-DLS with external interfaces power supply, BIP (a Russian acronym for the block of interfaces and memory), JTAG (Joint Test Action Group) and representative dimension; (b) The M-DLS view in its operating position on the landing platform with the MGAP (Martian Gas Analytical Package) interface.
Lines 124-126: Figure 3. Principal Martian air sample preparation system diagram, joint for both the M-DLS and the MGAP (Martian Gas Analytical Package) instruments. The air sample regulation (ASR) valve — air sample regulation valve.
Lines 290-293: Figure 6. Block diagram of electric circuits of the M-DLS. There are four main blocks: The Atmospheric Measurements Unit (AMU) for experiment handling; the Service System Unit (SSU) for auxiliary systems operation and data transfer to the landing platform; the air sample preparation unit and the ICOS cell for spectroscopic measurements.
Lines 364-365: Figure 7. Real-time line center search algorithm visualization. (a) — simulation of photodiode data for algorithm verification;
Lines 386-387: Figure 8. Principle scheme (a) and internal layout (b) of the M-DLS. SSU - Service System Unit, AMU - Atmospheric Measurements Unit.
- The ICOS performance will be significantly affected by the reflectivity of the mirrors of the gas cell. The effective absorption path length is determined by the reflectivity. It is possible that the mirror can be contaminated during the 1-year test. Can the authors comment on this issue? Then how to calibrate the instrument?
Yes, we expect the mirrors to be slowly affected by the martian dust. However, main scientific goal of the experiment is isotopologues ratios measurement, rather than absolute abundances. As we measure all spectral lines of interest, including those of main isotope and minor ones, within a single laser frequency scan, a change of the effective optical pass is not a problem. Moreover, the main isotopologue of CO2 is quite a stable component of the Martian atmosphere, and its lines present in both spectral ranges, this fact, along with the accurate measurement of pressure with the precision of 0.2%, allows us to calibrate the measurements with the required accuracy.
Corresponding sentences are added to the Discussion:
Lines 525-533: During the mission, the ICOS cell mirrors are expected to be gradually contaminated by the Martian atmospheric dust, with their reflectivity being degraded and the cell’s effective optical path shrunken. However, main scientific goal of the experiment is isotopologues ratios measurement, rather than absolute abundances. As we measure all spectral lines of interest, including those of main isotope and minor ones, within a single laser frequency scan, a change of the effective optical pass is not a problem. Moreover, the main isotopologue of CO2 is quite a stable component of the Martian atmosphere, and its lines present in both spectral ranges, this fact, along with the accurate measurement of pressure with the precision of 0.2%, allows us to calibrate the measurements with the required accuracy.
- The instrument has a power consumption of 15 W. Can the authors comment about the power supply? Battery integrated with solar panel?
Power supply for the instrument, as well as for the whole science payload of the landing platform, is provided by the centralized DC-DC converter of the spacecraft that maintains necessary current level without voltage spikes. M-DLS’ entrance voltage level is 28V, which is converted into 5V, and 15V by the internal electronics. As it was indicated on page 9 lines 341-344.
- It will be an impressive work if the proposed instrument can be used in the future space exploration.
Yes, we expect such an opportunity. For future development of the method, we plan to consider ICOS cells with longer effective optical path and optimized raytracing inside the cell to damp the interference. These measures may potentially increase the instrument’s sensitivity at least by one order of magnitude. Also, we plan to employ ICOS cell as a recedence channel for spaceborne NIR heterodyne laser spectroradiometer, which will be described elsewhere.
Corresponding sentences are added to the Discussion.
Lines 544-548: For future development of the method, we plan to consider ICOS cells with longer effective optical path and optimized raytracing inside the cell to damp the interference. These measures may potentially increase the instrument’s sensitivity at least by one order of magnitude. Also, we plan to employ ICOS cell as a recedence channel for spaceborne NIR multichannel heterodyne laser spectroradiometer [15], which will be described elsewhere.
